# Parental Decision Making Regarding COVID-19 Vaccines for Children under Age 5: Does Decision Self-Efficacy Play a Role?

**DOI:** 10.3390/vaccines11020478

**Published:** 2023-02-18

**Authors:** Jennifer D. Allen, Masako Matsunaga, Eunjung Lim, Gregory D. Zimet, Kimberly H. Nguyen, Holly B. Fontenot

**Affiliations:** 1Department of Community Health, Tufts University School of Arts and Sciences, 574 Boston Ave, Medford, MA 02155, USA; 2Department of Quantitative Health Sciences, John A. Burns School of Medicine, University of Hawaii at Manoa, Honolulu, HI 96813, USA; 3Department of Pediatrics, Indiana University School of Medicine, 410 West 10th Street, Suite 1001, Indianapolis, IN 46202, USA; 4Department of Public Health and Community Medicine, Tufts University Medical School, 136 Harrison Ave, Boston, MA 02111, USA; 5Nancy Atmospera-Walch School of Nursing, University of Hawaii at Manoa, 2528 McCarthy Mall, Webster Hall, Honolulu, HI 96822, USA

**Keywords:** SARS-CoV-2 (COVID-19), vaccination, parents, vaccine confidence, health behaviors, decision self-efficacy

## Abstract

Background: COVID-19 vaccines are now available under Emergency Use Authorization for children ages 6 months to 5 years. We examined parents’ intentions to vaccinate their children under the age of 5 years and assessed whether their confidence in making an informed decision about vaccination (decision self-efficacy) was associated with these intentions. Method: We conducted a cross-sectional online survey of U.S. parents between 23 March and 5 April 2022. We examined associations between parental intention to vaccinate their young children (<age 5 years) and confidence in vaccine decision making (decision self-efficacy). A multivariable multinomial logistic regression model was used to obtain adjusted odds ratios (AORs) and 95% confidence intervals (CIs) of parental intention (categorized as intend to vaccinate, unsure, or do not intend to vaccinate). Results: Of the 591 parents in this sample, 49% indicated that they intended to vaccinate their child(ren), 29% reported that they would not, and 21% were undecided. In bivariate analyses, race/ethnicity, health insurance, flu vaccination in the past 12 months, and parental COVID-19 vaccination status were significantly related to parental intention to vaccinate their child(ren). In the multivariable analyses, which controlled for these factors, parents who intended to vaccinate their child(ren) had greater confidence in their ability to make informed decisions about COVID-19 vaccinations compared to those who were unsure about vaccination. Each one standard deviation in the Decision Self-Efficacy score was associated with a 39% increase in intention to vaccinate one’s child versus being unsure about vaccination (AOR 1.39, 95% CI 1.09, 1.77). Conclusions: Parents who are unsure about vaccinating their children against COVID-19 may benefit from interventions designed to increase their ability to obtain, understand, and utilize information to make informed decisions.

## 1. Introduction

To date, over 15 million Coronavirus 2019 (COVID-19; SARS-CoV-2) cases in U.S. children have been reported [1], and there have been over 165,000 hospitalizations [2] and more than 1500 deaths in the U.S. [3]. While children may have similar or higher incidence rates of COVID-19 infection compared with adults, they more frequently experience asymptomatic infections or display a less severe illness [4,5]. However, while rare, some children may develop Multisystem Inflammatory Syndrome [6]. Additionally, long COVID-19 in children has been reported [7].

The Food and Drug Administration (FDA) approved the Pfizer-BioNTech and Moderna vaccines against COVID-19 under Emergency Use Authorization (EUA) for children ages 6 months through 5 years in June 2022 [8], and the Center for Disease Control and Prevention (CDC) Advisory Committee on Immunization Practice recommends routine COVID-19 vaccination for all children in this age group [9]. The availability of COVID-19 vaccines for young children is instrumental in protecting them from potential adverse consequences of infection and may help slow the spread of the virus in the community. However, according to a report from December 2022, only 11% of parents of children ages 6 months through 4 years reported that their child in this age group had been vaccinated for COVID-19 [10]. 

In general, prior studies have found that parents who are willing to vaccinate their children are older, male, more educated, and vaccinated against COVID-19 themselves [11,12,13]. Moreover, COVID-19 vaccine decisions have been strongly associated with political party affiliation [14,15]. Across studies, parents’ primary concerns about vaccinating their children include concerns about vaccine efficacy, side effects, and potential long-term health impacts [12,16,17,18]. Other factors associated with parental COVID-19 vaccine hesitancy include concerns that the vaccine development and testing have been rushed [19], distrust of the government [20], and a belief that vaccination is not needed because COVID-19 is less severe in children [20,21].

Research in this area is expanding rapidly. Nonetheless, a deeper understanding of how parents obtain, understand, and utilize information about COVID-19 vaccines is needed, particularly in the context of rampant mis- and disinformation about COVID-19 circulating on social media [22], inconsistent or unclear messaging from the CDC [23] and political leaders [24], and growing mistrust of the government [25]. Moreover, given the disproportionate COVID-19 burden on Black, Hispanic, and Asian communities [26], there is a particular need to understand and address factors influencing parental vaccine decisions among these groups. Our goal was to examine the decisional processes associated with parents’ intentions to vaccinate their young child(ren) (<5 years of age) among a racially and ethnically diverse sample in the U.S. We chose to focus on children under age 5 since research shows that parents have more concerns about vaccinating younger compared with older children [21].

Our study was guided by the Ottawa Decision Support Framework (ODSF), which integrates concepts from social-behavioral theories (e.g., the Health Belief Model, the Theory of Planned Behavior); decisional conflict theory; and expectancy-value theory [27]. The ODSF describes factors that influence individuals’ ability to make informed medical decisions, including knowledge; attitudinal factors (e.g., perceived barriers); and social pressure. A key factor required to make informed decisions is having the confidence to obtain, interpret, and act on health information, a construct known as “decision self-efficacy”. According to the framework, individuals who have adequate knowledge, positive attitudes toward a behavior, the ability to resist social pressure, and higher decision self-efficacy are better prepared to make high-quality decisions. 

## 2. Materials and Methods

We conducted a 15 min online survey (23 March to 5 April 2022) as part of a larger study of vaccine attitudes. Those eligible to participate in the larger study were aged 18 years or older, parents of child(ren) under the age of 18, able to read English, and members of a national Qualtrics panel. Qualtrics is a U.S. survey company that conducts health, social, and marketing research. In addition to being a cloud-based subscription software platform for creating and distributing surveys, Qualtrics maintains a “panel” or a pool of verified individuals designed to be representative of the U.S. population. Panelists have agreed to be contacted about participation in research. When research opportunities are available, invitations are sent via email with a hyperlink to take the survey in exchange for a modest incentive (USD 1). Informed consent information is presented on the first page of the survey, and individuals are unable to progress to survey questions prior to reading consent information and checking a box stating that they consent to study participation. We selected Qualtrics because a comparison of online U.S. survey panels (Facebook, MTurk) found that Qualtrics was the most demographically representative of the survey panels [28]. Quota sampling was used to ensure a balanced sample of 800 parents regarding gender and race/ethnicity (25% Black, 25% Latino, 25% Asian, and 25% White). Data for this analysis were restricted to parents of children less than 5 years of age, as our goal for this study was to assess parental attitudes toward vaccinating their youngest children. All study procedures were approved by the Institutional Review Board at Tufts University (protocol 00001954).

Outcome: Parental intention to vaccinate their child(ren), was assessed with one question: “If recommended for your child(ren)’s age group, would you have your child vaccinated for COVID-19?” Respondents could select from the following response options: “I plan to get them vaccinated”; “No, I don’t plan to get my child(ren) vaccinated”; or “I am unsure”. These were categorized as “intend to vaccinate”; “do not intend to vaccinate”; and “unsure about vaccination”, respectively. 

Independent variable: Decision Self-Efficacy was assessed via a modified 11-item Decision Self-Efficacy Scale [29]. The original scale was created to assess decisions regarding medications. For the current study, items were modified such that they inquired specifically about vaccine decisions (see Appendix A, Vaccine Decision Self-Efficacy Questions). For example, the original item regarding confidence in one’s ability to “Get the facts about the benefits of medications” was modified to confidence in one’s ability to make informed decisions about vaccination. Items assessed whether individuals felt confident that they could: “Get the facts about the benefits of vaccines”, “Get the facts about the risks of vaccines”, and “Handle unwanted pressure from others when making your choice about vaccination”. Response options included: “Not at all confident = 0”, “Somewhat confident = 2”, and “Very confident = 4”. The total score was rescaled to 100, with higher scores indicating greater decision self-efficacy. This scale showed excellent internal consistency (Cronbach’s alpha = 0.91).

Covariates: Items to assess socio-demographic and health characteristics were taken from the U.S. Behavioral Risk Factor Surveillance System, a nationwide system of health-related telephone surveys that collect state data on U.S. residents regarding their health-related risk behaviors, chronic health conditions, and use of preventive services [30]. We utilized these standardized items to enable cross-comparison with other U.S.-based studies. Respondent age was categorized according to the tertile of the 591 survey participants (27–32, 33–38, and 39–45 years old). Race/ethnicity was categorized as Hispanic, Non-Hispanic (NH) Asian, NH Black, NH White, and NH other race/multi-racial. Annual income was categorized as USD 75,000 or more; USD 50,000–74,999; USD 35,000–49,999; less than 35,000; and unsure. Primary source of healthcare coverage was categorized as insured and uninsured/do not know. Children’s flu vaccine was dichotomized (yes/no) after asking parents if they had had their child vaccinated for flu in the past 12 months. Those who selected “No, I don’t plan to get my child vaccinated for the flu this year” or “I’m not sure” were categorized as “No”. Those who selected “Yes, my child has already received the flu vaccine in the past 12 months” or “Yes, I plan to get my child vaccinated for the flu this year” were categorized as “Yes”. We also inquired about the parent’s COVID-19 vaccination status, asking: “Have you received all required doses of the COVID-19 vaccine?” This referred to initial vaccination and did not assess the receipt of booster doses. Individuals could respond: “Yes, I received all required doses”; “Yes, I plan to receive all required doses”; or “No, I don’t plan to receive all required doses”. Those who responded in the affirmative were categorized as “Vaccinated/plan to be vaccinated” versus “Not vaccinated/do not plan to be vaccinated”.

### Statistical Analysis

Of the total of 828 parents surveyed, 591 (71.4% of the total) had children aged under five years of age, representing the sample included in these analyses. Parental socio-demographic characteristics were described in frequencies and percentages or means and standard deviations (SDs). Differences in socio-demographic and health characteristics across the outcome (parental intention to vaccinate their child) were examined using the Pearson Chi-square test or Fisher’s exact test for categorical variables and analysis of variance for continuous variables. A multinomial logistic regression model compared the odds of parents who were unsure about vaccinating their child(ren) to those who did intend to vaccinate and those who did not intend to vaccinate. Variables with a *p*-value less than 0.20 in these analyses were included in the model to explore associations between parental intention to vaccinate their child(ren) with a one-standard-deviation change in Decision Self-Efficacy scores. We designated parents who reported that they were “unsure” about vaccinating their child(ren) as the referent group, since this group was likely to be more amenable to vaccine interventions than those who did not intend to vaccinate [31,32]. Adjusted odds ratios (AORs) and 95% confidence intervals (CIs) were obtained to evaluate associations. The AOR for Decision Self-Efficacy score was reported for one SD of change controlling for the covariates. A *p*-value less than 0.05 was considered statistically significant. 

## 3. Results

Table 1 shows the characteristics of the parents in the study. The mean age was 35.5 years (SD = 5.1). There was a relative balance in terms of gender and race/ethnicity across the sample. Most (85.6%) reported that they had health insurance. More than half (59.4%) reported that their child(ren) had been vaccinated against the flu in the past year. In terms of parental COVID-19 vaccination status, 71.7% reported that they had been vaccinated or planned to be fully vaccinated. About half (49.1%) of parents reported that they intended to vaccinate their child(ren) for COVID-19, 21.5% were unsure, and 29.4% did not intend to vaccinate their child(ren). 

Men were more likely to report that they intended to vaccinate their child(ren) or to be unsure about vaccination compared with women (*p* = 0.016). Race/ethnicity was also significantly associated with parental intentions (*p* < 0.001); among those who intended to vaccinate their child(ren), 27.2% were NH White, 21.4% were NH Asian, 24.1% were Hispanic, 20.3% were NH Black, and 6.9% were NH other/multi-racial. Having a primary source of healthcare (*p* = 0.009) and child flu vaccination status (*p* < 0.001) were significantly associated with parental intentions to vaccinate their child(ren). The results showed that the Decision Self-Efficacy Scale scores varied across the three groups (*p* < 0.001, *η*^2^ = 0.05). Parents who intended to vaccinate their children had the highest mean scores (74.3, SD = 21.8), followed by those who did not intend to vaccinate their children (68.6, SD = 25.2) and those who were unsure (64.1, SD = 23.8).

Table 2 presents the AORs and 95% CIs for the associations between parental intentions to vaccinate their child(ren) in terms of a one-standard-deviation change in the Decision Self-Efficacy score. This model controlled for the variables with a *p*-value less than 0.20 found in the bivariate analysis (parent age, gender, race/ethnicity, health insurance coverage, child’s flu vaccination status, and parent COVID-19 vaccination status); each standard deviation in the Decision Self-Efficacy scale was associated with a 39% increase in the odds of intending to vaccinate one’s child(ren) compared with those who were unsure about vaccinating their child(ren) (AOR 1.39, 95% CI 1.09, 1.77). However, Decision Self-Efficacy scores were not significantly different among those who did not intend to vaccinate their child(ren) and those who were unsure. 

Other factors that distinguished those who were unsure versus those who did not intend to vaccinate their child(ren) were being female, NH Asian, NH other race/multi-racial, child having been vaccinated for flu in the past year, and parental COVID-19 vaccination status. Most of these characteristics also distinguished those who were unsure from those who did intend to vaccinate their child(ren), with the exception of gender. See Table 2.

## 4. Discussion

In this sample of racially/ethnically diverse mothers and fathers, we found that approximately half (49%) intended to vaccinate their child(ren) under the age of 5 years, while 22% were unsure, and 29% did not intend to vaccinate their child(ren). After adjusting for parental and child characteristics known to be associated with COVID-19 intentions or behaviors, we found that parents who intended to vaccinate their child(ren) had greater confidence in their ability to make informed decisions about COVID-19 vaccinations compared to those who were unsure. 

Our finding that about half of the parents intended to vaccinate their children is consistent with a study conducted in February 2022 among a diverse sample of parents of children ages 4 months through 4 years, which reported that approximately 50% of parents intended to vaccinate their child(ren) “at some point”, although only one in five parents (19%) stated that they would do so within 3 months of their child’s eligibility to receive a vaccination [33]. Similarly, a poll taken by the Kaiser Family Foundation in April 2022 found that 18% of parents intended to vaccinate their child “right away,” 38% would “wait and see,” 11% would have their child vaccinated only if required, and 27% would definitely not vaccinate their child [21]. In an earlier study using the Pediatric Research Observing Trends and Exposures in COVID-19 Timelines survey, the vaccine attitudes of 393 parents of children less than 4 years of age across four states were examined. The findings revealed that 64% of parents were likely to vaccinate their child(ren), 19% were unsure, and 10% were unlikely to have their child aged <5 years receive the COVID-19 vaccine. Notably, at three months follow-up, parents overall were less likely to report that they would vaccinate their child(ren) and had higher levels of mistrust in the government than they did at baseline [12].

To our knowledge, this was the first study to examine the role of decision self-efficacy in the context of COVID-19 vaccination. While numerous studies have examined parental beliefs about vaccine efficacy [19,34] or self-efficacy with regard to the ability to overcome barriers to vaccination [35], we were unable to identify any study that has examined decision self-efficacy among adults or parents related to COVID-19. 

Before discussing study implications, we acknowledge its limitations. First, this was a non-probability sample among a Qualtrics study panel, so caution is needed when generalizing findings. While electronically recruited panels are understood to not completely represent the U.S. population, they are equally as representative as traditional recruitment approaches [36]. Second, due to the cross-sectional nature of the data collection, we could not draw causal inferences or rule out reciprocal causation. Ideally, future studies will utilize prospective data to investigate factors that predict parental intentions to vaccinate their children. Third, at the time of data collection, no COVID-19 vaccines had been approved for children under age 5 years. Additionally, we could not exclude parents with children less than 6 months of age, because child age was only collected in years. Parents with infants may be even more hesitant about vaccination than parents of older children [37]. Fourth, additional factors are known to impact COVID-19 vaccine decisions, such as political party affiliation, knowledge of COVID-19 vaccines, and the endorsement of naturalistic and homeopathic approaches to prevention [38]. We did not assess these factors for multiple reasons, including the fact that these issues have been well-studied already and that political affiliation is not a reasonable target for behavioral interventions to improve vaccination rates, and the desire to minimize respondent burden. Fifth, a power analysis was not conducted prior to the study. However, the post-study power analysis showed that our sample size of 591 parents was sufficient to examine the associations between decision self-efficacy and parental decisions about vaccinating their children against COVID-19, with a power greater than 99%. Unlike much of the existing research on COVID-19 vaccine attitudes, which has utilized large-scale surveys that limit the ability to assess latent theoretical constructs using validated instruments, a strength of our study was the in-depth exploration of a validated measure of confidence in decision making. Notably, we believe this to be the first study to assess parental decision self-efficacy regarding COVID-19 vaccines. Additional study strengths include the racial/ethnic diversity of the sample and the timeliness of the findings.

Despite these limitations, we believe that this study provides new and important information about decision-making processes among parents who are unsure about having their young children vaccinated against COVID-19. In our sample, nearly a quarter of the parents were unsure about vaccinating their child(ren). This is an important target group for vaccination campaigns, since those who are uncertain are more amenable to interventions than those who outright refuse vaccines.

## 5. Conclusions

The current study showed that parents who intended to vaccinate their child(ren) aged under 5 years were confident in their ability to make informed decisions about COVID-19 vaccinations compared to those who were unsure. This indicated that those who are unsure about vaccinating their child(ren) against COVID-19 may benefit from interventions designed to increase their ability to obtain, understand, and utilize information to make informed decisions. Prior to the pandemic, interventions to support decision making among vaccine-hesitant parents focused on patient education [39], encouraging healthcare providers to use ‘presumptive’ language in vaccine recommendations [40], and addressing environmental factors, including social norms and social media influences [41]. These strategies remain central in the context of COVID-19 and are in line with the CDC’s “Vaccinate with Confidence” [42] recommendations, which emphasize the need for messaging that is clear and consistent, addresses parents’ primary concerns, and proactively addresses misinformation. These results also align with the CDC’s strategies for increasing COVID-19 vaccination through motivational interviewing, i.e., a conversation between health professionals and patients that can influence patients’ willingness to receive a vaccination [43]. When healthcare providers speak with unvaccinated patients about vaccination in an evidence-based and culturally sensitive way, patients can manage their feelings and move toward healthy behavior changes. This process suggests that motivational interviewing can be used to help patients improve their self-efficacy and decision making regarding vaccination. 

Our findings suggested that one intervention approach for parents who are unsure about vaccination might be to improve their confidence in COVID-19 vaccine decision making. This could be achieved by providing clear information about vaccine safety and efficacy that is delivered in a way that enhances parents’ sense of agency. For example, decision support interventions such as decision aids aim to support patient decision making by making decisions explicit and providing information about the options and the associated benefits and harms of a course of action [44]; these have been shown to be helpful in aiding parental decision making about other childhood vaccines [45]. Moreover, a patient-centered approach employing principles of motivational interviewing may also be an effective method to engender decisional self-efficacy for parents making decisions about vaccination [46], even with increased vaccine skepticism. Finally, since a majority of parents continue to look to their child’s healthcare provider for accurate and trustworthy information on vaccination to aid in decision making [47], healthcare providers have an important role in promoting confidence in COVID-19 vaccine decision making among parents of young children [48].

## Figures and Tables

**Table 1 vaccines-11-00478-t001:** Parent characteristics by intention to vaccinate child(ren) < 5 years for COVID-19, United States, 23 March to 4 April 2022, n = 591.

Characteristics	Total n = 591	Intend to Vaccinate Child n = 290 (49%)	Unsure about Vaccinating Child n = 127 (22%)	Do Not Intend to Vaccinate Child n = 174 (29%)	*p*-Value ^1^
Age (mean = 35.1, SD = 5.1), n (%)					
27–32 y	210 (35.5)	103 (35.5)	45 (35.4)	62 (35.6)	0.19
33–38 y	205 (34.7)	95 (32.8)	39 (30.7)	71 (40.8)	
39–45 y	176 (29.8)	92 (31.7)	43 (33.9)	41 (23.6)	
Gender, n (%)					
Male	286 (48.4)	147 (50.7)	70 (55.1)	69 (39.7)	0.016
Female	305 (51.6)	143 (49.3)	57 (44.9)	105 (60.3)	
Race/Ethnicity, n (%)					<0.001
Hispanic	149 (25.2)	70 (24.1)	31 (24.4)	48 (27.6)	
NH Asian	115 (19.5)	62 (21.4)	32 (25.2)	21 (12.1)	
NH Black	131 (22.2)	59 (20.3)	24 (18.9)	48 (27.6)	
NH White	131 (22.2)	79 (27.2)	19 (15.0)	33 (19.0)	
NH other race or multi-racial	65 (11.0)	20 (6.9)	21 (16.5)	24 (13.8)	
Annual Household Income, n (%)					
USD 75,000 or more	187 (31.6)	105 (36.2)	36 (28.3)	46 (26.4)	0.20
USD 50,000–74,999	139 (23.5)	70 (24.1)	32 (25.2)	37 (21.3)	
USD 35,000–49,999	91 (15.4)	39 (13.4)	20 (15.7)	32 (18.4)	
Less than USD 35,000	165 (27.9)	71 (24.5)	36 (28.3)	58 (33.3)	
Unsure	9 (1.5)	5 (1.7)	3 (2.4)	1 (0.60)	
Health Insurance Coverage, n (%)					0.009
Insured	506 (85.6)	260 (89.7)	108 (85.0)	138 (79.3)	
Uninsured/Do not know	85 (14.4)	30 (10.3)	19 (15.0)	36 (20.7)	
Child flu vaccination in past 12 months, n (%)					<0.001
No	240 (40.6)	71 (24.5)	57 (44.9)	112 (64.4)	
Yes	351 (59.4)	219 (75.5)	70 (55.1)	62 (35.6)	
Parental COVID-19 vaccination, n (%)					<0.001
Vaccinated/plan to vaccinate	424 (71.7)	270 (93.1)	86 (67.7)	68 (39.1)	
Not vaccinated/do not plan to vaccinate	167 (28.3)	20 (6.9)	41 (32.3)	106 (60.9)	
Decision Self-Efficacy Scale					
Mean (SD)	68.9 (23.8)	74.3 (21.8)	64.1 (23.8)	68.6 (25.2)	<0.001
Range	0, 100	5, 100	0, 100	0, 100	

NH = Non-Hispanic; ^1^ A *p*-value was computed from Chi-square test or Fisher’s exact test for categorical variables and analysis of variance for continuous variables (For annual household income, a simulated *p*-value was computed by Fisher’s exact test with 2000 replicates).

**Table 2 vaccines-11-00478-t002:** Adjusted odds ratios and 95% confidence intervals for parental intention to vaccinate child(ren) for COVID-19 and decision self-efficacy, United States, 23 March to 4 April 2022, n = 591.

	Do Not Intend to Vaccinate vs. Unsure		Intend to Vaccinate vs. Unsure	
	AOR	95% CI	AOR	95% CI
Decision Self-Efficacy ^1^				
	1.10	0.86, 1.41	1.39	1.09. 1.77
Age group				
27–32 y	1.00	Ref.	1.00	Ref.
33–38 y	1.54	0.86, 2.76	1.07	0.62, 1.86
39–45 y	0.82	0.44, 1.53	1.07	0.62, 1.86
Race/Ethnicity				
NH White	1.00	Ref.	1.00	Ref.
Hispanic	0.66	0.30, 1.45	0.54	0.26, 1.08
NH Black	0.74	0.33, 1.67	0.61	0.29, 1.29
NH Asian	0.36	0.16, 0.83	0.39	0.19, 0.78
NH another race or multi-racial	0.34	0.14, 0.83	0.27	0.11, 0.65
Gender				
Male	1.00	Ref.	1.00	Ref.
Female	1.81	1.08, 3.02	1.41	0.88, 2.24
Healthcare Coverage				
Insured	1.00	Ref.	1.00	Ref.
Uninsured/do not know	1.32	0.68, 2.56	1.07	0.54, 2.14
Child flu vaccination in past 12 months				
No	1.00	Ref.	1.00	Ref.
Yes	0.50	0.31, 0.83	1.94	1.21, 3.09
Parent COVID-19 Vaccination				
Not vaccinated	1.0	Ref.	1.0	Ref.
Vaccinated	0.33	0.19, 0.56	4.73	2.52, 8.86

AOR = adjusted odds ratio. CI = confidence interval. Ref. = reference. ^1^ AORs reflect a one-standard-deviation increase in Decision Self-Efficacy score controlling for age group, race/ethnicity, gender, healthcare coverage, child flu vaccination, and parental COVID-19 vaccination.

## Data Availability

The data presented in this study are openly available in OSF repository, under Allen, J. D. (17 February 2023). Parental decision-making about COVID-19 vaccines for children under age 5: Does Decision Self-Efficacy play a role? Retrieved from Appendix A.

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
