# Peer review of "Parental Decision Making Regarding COVID-19 Vaccines for Children under Age 5: Does Decision Self-Efficacy Play a Role?"

_vaccines, 2023, doi:10.3390/vaccines11020478_

Round 1

Reviewer 1 Report

This is an interesting article for people in the field and others researching. It's good for planning for govt agencies, insurance companies etc. 

Author Response

This is an interesting article for people in the field and others researching. It's good for planning for government agencies, insurance companies etc. 

Author Response: Thank you for this feedback.

Reviewer 2 Report

The question of whether parents would choose to vaccinate their children against SARS-CoV-2, and why they might not do so, is important.  Unfortunately, this evaluation did not consider many relevant factors, which would also determine those decisions beyond a sense of self-efficacy.

For example, the subject of vaccination against COVID became very politicized.  It is likely that a support of democrats would be more likely to decide to vaccinate their children than a republican.  Political affiliation was not assessed.

There is also a strong influence of SES, which would not only affect decisions about health, but also one's sense of efficacy and empowerment.

Beyond left- and right-wing politics, there are many who endorse more naturalistic and homeopathic approaches and are fearful of modern medicine. Even among clinical professionals.  For example, surveys indicate that about 50% of chiropractors do not believe in vaccines.

Decisions about vaccines have also changed dramatically over time.  Even many who opted to get a primary immunization then chose not to get a booster.  Many seem to be un-informed or mis-informed about the difference between the initial Pfizer and Moderna vaccine and the later bivalent formulation.

It would have been of value to inquire about who much the participants knew about SARS-CoV-2, its capacity for mutation, and the benefits of immunization.  For example, the difference between stimulating protective immunity vs. sterilizing immunity.

It would also have been of value to know more about whether the parents thought the vaccines were not necessary for their children or potentially dangerous with likely side effects.

In sum, the conclusions are very limited by what seems to have been a superficial survey instrument.  At this point, the only remedy would be to have a very detailed limitations and caveat section in Discussion, which acknowledged all of these concerns.

The N is also fairly small for this kind of analysis despite what might seem to be a large enough number to be representative of a larger population.

Author Response

The question of whether parents would choose to vaccinate their children against SARS-CoV-2, and why they might not do so, is important.  Unfortunately, this evaluation did not consider many relevant factors, which would also determine those decisions beyond a sense of self-efficacy.

1.    For example, the subject of vaccination against COVID became very politicized.  It is likely that a support of democrats would be more likely to decide to vaccinate their children than a republican.  Political affiliation was not assessed.

Author Response: You are correct in that we did not assess political party affiliation in this study. You are also correct in that there is abundant evidence that COVID-19 vaccination has been highly politicized and that COVID-19 vaccine behaviors differ by political party affiliation in the U.S. Given the voluminous literature documenting this relationship, we did not assess political party affiliation in our survey. Rather, the goal of this theory-based study was to focus primarily on factors for which there has been little prior evidence. Furthermore, it is likely that political party affiliation, in and of itself, is not the key issue. Instead, political affiliation almost certainly is mediated in its impact on COVID-19 vaccination via health beliefs and attitudes. However, per your suggestion, we have added information to the introduction section about the politicization of COVID-19 vaccines (see lines 95-96) and have cited the absence of information about political party affiliation as a limitation in the discussion section.

Introduction
 “Moreover, vaccine decisions among adults have been strongly associated with political party affiliation.1,2”

Discussion (see lines 479- 482)
“… there are additional factors that are known to impact COVID-19 vaccine decisions, such as political party affiliation, knowledge of COVID-19 vaccines, as well as the endorsement of naturalistic and homeopathic approaches to prevention.39 We did not assess these factors for multiple reasons, including the fact that these issues have been well-studied already, political affiliation is not a reasonable target for behavioral interventions to improve vaccination rates, and the desire to minimize respondent burden.”

2.    There is also a strong influence of SES, which would not only affect decisions about health, but also one's sense of efficacy and empowerment.

Author Response: This is a valid comment, as there is indeed solid evidence to demonstrate that COVID-19 vaccination behaviors are associated with measures of socio-economic status, primarily income. The fact that we did not observe a statistically significant association between income and vaccine intentions in our study (p<0.20) was unexpected. 

It has been hypothesized that the mechanism underlying the relationship between SES and COVID-19 vaccination is that those with higher SES have a greater sense of efficacy and empowerment when it comes to COVID-19 vaccination. However, we have not identified any empirical studies that explicitly examined decision self-efficacy (or empowerment) and parental intentions to vaccinate their children for COVID-19. As such, we believe that our study provides important new insights. 

3.    Beyond left- and right-wing politics, there are many who endorse more naturalistic and homeopathic approaches and are fearful of modern medicine. Even among clinical professionals.  For example, surveys indicate that about 50% of chiropractors do not believe in vaccines.

Author Response: This is a valid point. However, given that this was not our primary research question, we did not assess the relationship between the endorsement of naturalistic and homeopathic approaches, we are unable to comment on this in our paper. However, we have noted this in the discussion section under limitations (see lines 479- 482).

Discussion
“… there are additional factors that are known to impact COVID-19 vaccine decisions, such as political party affiliation, knowledge of COVID-19 vaccines, as well as the endorsement of naturalistic and homeopathic approaches to prevention.39 We did not assess these issues for multiple reasons, including the fact that these factors has been well-studied already, political affiliation is not a reasonable target for interventions to improve vaccination rates, and the desire to minimize respondent burden.”

4.    Decisions about vaccines have also changed dramatically over time.  Even many who opted to get a primary immunization then chose not to get a booster.  Many seem to be un-informed or mis-informed about the difference between the initial Pfizer and Moderna vaccine and the later bivalent formulation.

Author Response: Indeed, beliefs about COVID-19 vaccines have changed over time and there does appear to be confusion about various vaccine formulations. Since our study was cross-sectional, we are unable to comment on trends over time. Of note is that the bivalent vaccines were not available at the time our survey was conducted (March 23 and April 5, 2022) and therefore, confusion between single- and bivalent- vaccines is unlikely to have impacted our findings. 

5.    It would have been of value to inquire about how much the participants knew about SARS-CoV-2, its capacity for mutation, and the benefits of immunization.  For example, the difference between stimulating protective immunity vs. sterilizing immunity.

Author Response: We can see the value of that research question. However, we did not assess knowledge of SARS-CoV-2, its’ ability to mutate, or the difference between protective vs sterilizing immunity, so are unable to comment on this line of inquiry. Our theory-informed study goal was to assess whether parents felt confident in their ability to obtain, interpret and make decisions on information about the COVID-19 vaccines (regardless of whether their interpretation was correct). As suggested, we note this as a limitation (see lines XX)

Discussion (see lines 479- 482)
“… there are additional factors that are known to impact COVID-19 vaccine decisions, such as political party affiliation, knowledge of COVID-19 vaccines, as well as the endorsement of naturalistic and homeopathic approaches to prevention.39 We were unable to assess these factors given budgetary limitations and the desire to minimize respondent burden.”

6.    It would also have been of value to know more about whether the parents thought the vaccines were not necessary for their children or potentially dangerous with likely side effects.

Author Response: Many studies have documented that parents' primary concerns about vaccinating their children against COVID-19 are based on potential side effects. We note this in the introduction, stating: “Across studies, parents’ primary concerns about vaccinating their children include concerns about vaccine efficacy, side effects, and potential long-term health impacts.12,16–18” (see lines 97-101). Our goal for this study was to examine the latent construct of confidence in decision-making, whether based on accurate or inaccurate information.

7.    In sum, the conclusions are very limited by what seems to have been a superficial survey instrument.  At this point, the only remedy would be to have a very detailed limitations and caveat section in the Discussion, which acknowledged all of these concerns.

Author Response: Thank you for this suggestion. As is noted above, we have added more information about study limitations to the Discussion section. Please 

8.    The N is also fairly small for this kind of analysis despite what might seem to be a large enough number to be representative of a larger population

Author Response: We do not claim that the study is representative of the entire U.S. population. To clarify, we have added the following language to the discussion section (see lines 469-493):

“First, this was a non-probability sample among a Qualtrics study panel, so caution is needed when generalizing findings. While electronically-recruited panels are understood not to be completely representative of the U.S. population, they are equivalently representative as traditional recruitment approaches.”

Reviewer 3 Report

Thank you for sharing your manuscript on parental decision-making for COVID-19 vaccination among young children. Here some comments/questions that could help to improve the article:

Introduction

-Does "there have been over 165000 hospitalisations, and more Han 1500 deaths" also relate to U.S. children?

-Rephrase "some children may develop Multisystem Inflammatory Syndrome and long COVID-19 in children has been reported" for more clarity.

-Correct "for children ages 6 months through 5 year". Also, is the targeted age range up to 4 years or 5 years of age? Be consistent throughout. 

Materials and Methods

-Were the parents enrolled informed about which brand of vaccine could be administered to their children? Did you stratify your assessment by vaccine, i.e., BioNTech versus Moderna, as willingness to vaccinate children may also depend on the vaccine brand itself? Also, when considering to vaccinate young children, would they receive a single or multiple doses? The overall aim of your manuscript really is important, but data collected and analyses performed inclusive of resulting findings would need to be presented with much more detail and clarity to draw conclusions.

-Did you perform a sample size calculation to assure your findings are statically sound and meaningful?   

-Please explain "Qualtrics" in more detail for readers unfamiliar with this platform. Why was this particular group/were these members selected? Could this have introduced any selection bias in your research also in terms of the sampling scheme chosen?

-Regarding the eligibility criteria, you stated that children had to be <18 years of age and further down in the same paragraph you selected children <5 years only for your analysis only. Please explain in more detail why you chose this approach.

-Consider describing your questions/variables including answer options in a table-like style rather than free text. The same applies to your scoring/rescaling. Describe your co-variates in detail rather than simply referring to behavioral risk factors of a surveillance system. 

-How did you obtain adjusted odds ratios? For what factors did you adjust the odds ratios for? This needs to be explained in the material & method section.

-Please rearrange variables shown in Table 1 for more clarity. 

-What type of informed consent did you obtain from parents, i.e., written or verbal? 

-As you enrolled participants from different races/ethnicities, were there any language barriers? If so, how did you deal with it also in terms of obtaining consent.

-Why did you assess past/previous flu vaccination only and not other vaccines?

-For the parents that stated "COVID-19 vaccinated", how many doses did they received and of which vaccine brand?  

Author Response

Thank you for sharing your manuscript on parental decision-making for COVID-19 vaccination among young children. Here are some comments/questions that could help to improve the article:

Introduction

1.    Does "there have been over 165000 hospitalizations, and more Han 1500 deaths" also relate to U.S. children?

Author Response: This statement does refer to children in the U.S. We have clarified this point. We have revised this statement to make this clear (see line 79):

“To date, over 15 million Coronavirus 2019 (COVID-19; SARS-CoV-2) cases in U.S. children have been reported,1 there have been over 165,000 hospitalizations,2 and more than 1,500 deaths in the U.S.3”

2.    Rephrase "some children may develop Multisystem Inflammatory Syndrome and long COVID-19 in children has been reported" for more clarity.

Author Response: Upon your suggestion, we have rephrased this sentence (see lines 81-82). It now states: 

“… while rare, some children may develop Multisystem Inflammatory Syndrome.6 Additionally, long COVID-19 in children has been reported.7”

3.    Correct "for children ages 6 months through 5 year". Also, is the targeted age range up to 4 years or 5 years of age? Be consistent throughout. 

Author Response: Thank you for pointing out this inconsistency. The study includes children under the age of 5. We have clarified this throughout the paper. 

Materials and Methods
4.    Were the parents enrolled informed about which brand of vaccine could be administered to their children? Did you stratify your assessment by vaccine, i.e., BioNTech versus Moderna, as willingness to vaccinate children may also depend on the vaccine brand itself? Also, when considering to vaccinate young children, would they receive a single or multiple doses? The overall aim of your manuscript really is important, but data collected and analyses performed inclusive of resulting findings would need to be presented with much more detail and clarity to draw conclusions.

Author Response: We did not assess the brand of vaccine that would be administered to children under age 5 because, at the time of data collection, it was unknown which vaccines would be approved by the FDA for this age group. Likewise, the number of required doses was also not known. It should be noted that in the U.S., there was little choice regarding vaccine brands for adults, and early recommendations for adults were to stick to the first vaccine brand for the second dose. 

5.    Did you perform a sample size calculation to assure your findings are statically sound and meaningful?   

Author Response: 
We did not perform a prior sample size calculation. Our sample of 591 adults is sufficient to examine associations between decision self-efficacy and parental decisions about vaccinating their children against COVID-19 with a power greater than 99%. 

6.    Please explain "Qualtrics" in more detail for readers unfamiliar with this platform. Why was this particular group/were these members selected? Could this have introduced any selection bias in your research also in terms of the sampling scheme chosen?

Author Response: We have added information about Qualtrics to the methods section (see lines 133-143). Specifically, the text now states:

“Qualtrics is a U.S. survey company that conducts health, social, and marketing research. In addition to being a cloud-based subscription software platform for creating and distributing surveys, Qualtrics maintains a “panel” or a pool of individuals designed to be representative of the U.S. population. Panelists are verified individuals who have agreed to be contacted about participation in research. When research opportunities are available, invitations are sent via email with a hyperlink to take the survey in exchange for a modest incentive. We selected Qualtrics because a comparison of most online U.S. survey panels (Facebook, MTurk) found that Qualtrics was the most demographically most representative of the survey panels.” 

We have also discussed potential limitations of the Qualtrics panel in the discussion section (see lines 469-493).

“First, this was a non-probability sample among a Qualtrics study panel, so caution is needed when generalizing findings. While electronically-recruited panels are understood not to be completely representative of the U.S. population, they are equivalently representative as traditional recruitment approaches.37”

7.    Regarding the eligibility criteria, you stated that children had to be <18 years of age and further down in the same paragraph you selected children <5 years only for your analysis only. Please explain in more detail why you chose this approach.

Author Response: We surveyed parents of youth under the age of 18 in a larger study but restricted the current analysis to parents of children under age 5 because parents tend to be most conservative about vaccinating infants, given concerns about the number of vaccines given during this period of development, as well as concerns that an infant’s immune system is not fully mature. Our goal for this study was to assess parental attitudes toward vaccinating children in the youngest age group.

8.    Consider describing your questions/variables including answer options in a table-like style rather than free text. The same applies to your scoring/rescaling. Describe your co-variates in detail rather than simply referring to behavioral risk factors of a surveillance system. 

Author Response: We have clarified that the Behavioral Risk Factor Surveillance System is a nationwide system of health-related telephone surveys that collect state data about U.S. residents regarding their health-related risk behaviors, chronic health conditions, and use of preventive services. We utilized standardized questions from this surveillance system so as to enable cross-comparison with other U.S.-based surveys, which also use these standardized questions. We have included an appendix with verbatim questions for the 11-item Decision Self-Efficacy measure. Details about the scoring of this variable are provided in the text (see lines 181-185).

“Items to assess socio-demographic and health characteristics were taken from the U.S. Behavioral Risk Factor Surveillance System, a nationwide system of health-related telephone surveys that collect state data about U.S. residents regarding their health-related risk behaviors, chronic health conditions, and use of preventive services.4 We utilized these standardized items to enable cross-comparison with other U.S.-based studies.”

9.    How did you obtain adjusted odds ratios? For what factors did you adjust the odds ratios for? This needs to be explained in the material & method section.

Author Response: Odds ratios were obtained through multinomial logistic regression models comparing those who were unsure about vaccination vs those who intended to vaccinate and those who did not. In the final model, we adjusted for factors that were significantly associated with parental intentions in bivariate analyses. In the methods section (see lines 211-214), we have clarified this stating:

“Multinomial logistic regression models compared the odds of parents who were unsure about vaccinating their children, compared to those who did intend to vaccinate and those who did not intend to vaccinate. Variables with a p-value less than 0.20 in these analyses were included in these models to explore associations (adjusted odds ratios) between parental intention to vaccinate their child(ren) with a one standard deviation change in Decision Self-Efficacy scores. We designated parents that reported that they were “unsure” about vaccinating their children as the referent group since this group is likely to be more amenable to vaccine interventions than those who do not intend to vaccinate.31,32 Adjusted odds ratios (AORs), and 95% confidence intervals (CIs) were obtained to evaluate associations. The AOR for Decision Self-Efficacy score was reported on one standard deviation (SD) of change controlling for covariates that remained significant at the p< 0.05 level (age group, race/ethnicity, gender, healthcare coverage, child flu vaccination, and parental COVID-19 vaccination).”

We also include information about factors adjusted in Table 2 in the footnote:

“AORs reflect one standard deviation increase in Decision Self-Efficacy score controlling for age group, race/ethnicity, gender, healthcare coverage, child flu vaccination, and parental COVID-19 vaccination.” 

10.    Please rearrange variables shown in Table 1 for more clarity. 

Author Response: We have utilized the standard format for Table 2 presenting the socio-demographic and health characteristics of the sample. All authors agree that the order of variables is presented according to most journal requirements. We can see that the copy-edit version of the manuscript created by the publisher has not been formatted correctly. Perhaps this is the reason for the suggestion? If not, please provide more detailed feedback about changes suggested for Table 2. 

11.    What type of informed consent did you obtain from parents, i.e., written or verbal? 

Author Response: Thank you for pointing out this omission. We have added information below (see lines 138-141)

“Informed consent information is presented on the first page of the survey and individuals are unable to progress to survey questions prior to reading consent information and checking a box stating that they consent to study participation.”

12.    As you enrolled participants from different races/ethnicities, were there any language barriers? If so, how did you deal with it also in terms of obtaining consent.

Author Response: We have clarified that eligibility requirements for study participation included the ability to read English. Please see lines 131-132

“Those eligible to participate were aged 18 years or older, parents of children under the age of 18, able to read English, and members of a national Qualtrics panel.”

13.    Why did you assess past/previous flu vaccination only and not other vaccines?

Author Response: Our primary research question was to assess parental attitudes toward COVID-19 vaccines. Also, like COVID-19 vaccination, flu vaccination is for prevention or mitigation of a respiratory infection and requires ongoing vaccination due to changes in circulating viral strains.

14.    For the parents that stated "COVID-19 vaccinated", how many doses did they received and of which vaccine brand?  

Author Response: To assess parental COVID-19 vaccination, we asked parents: “Have you received all required doses of the COVID-19 vaccine?” This referred to initial vaccination and did not assess receipt of booster doses. Individuals could respond: “Yes, I received all required doses,” “Yes, I plan to receive all required doses,” or “No, I don’t plan to receive all required doses.” Those who responded in the affirmative were categorized as “Vaccinated/plan to be vaccinated” versus “Not vaccinated/do not plan to be vaccinated. These are standard questions from the U.S. Center for Disease Control and Prevention Household Pulse Survey. 

We did not inquire about which vaccine (e.g., Pfizer, Moderna) received because, in the initial rollout of COVID-19 vaccines in the U.S., individuals did not have a choice of which brand of vaccine they would receive. For the second dose, at the time, individuals in the U.S. were advised to take the same brand of vaccine as the first dos

Round 2

Reviewer 2 Report

The authors have been responsive to the prior reviews and added a number of important caveats and limitations about the study methods and conclusions. 

My primary concerns about the experimental methods and design would have had to be addressed at the time of conducting the research and cannot be remedied after the fact.

Specifically, they just did not enough information to truly understand why some parents decided to vaccinate and others did not.  I would imagine that a sense of 'self-efficacy' was a factor, but it is not possible to determine its relative weight.  For example, relative to the a person's political affiliation.  Or time spent on the internet reading false information about vaccinations in general and mRNA vaccines in specific.  

For example, while information about a parent's decision to have the child vaccinated against flu in the prior year doesn't fully inform about whether parents opted to also not have other pediatric vaccines administered during the first year of life, such as DPT or against measles.

Had it been my study, I might also have included 1 or 2 items that probed whether the parents had even a rudimentary knowledge of what a virus is or how a vaccine primes the immune system to have a memory of the pathogen (but in a benign way).

Even with a focus on self-efficacy, we don't really know if the parents feel efficacious in other aspects of their lives.

Author Response

We appreciate the reviewers' comments and are delighted that reviewers 1 and 3 believe that the paper should be accepted. We understand Reviewer 2’s concerns and recognize that our study, like all other published research, has limitations. However, many of the reviewer's comments are not actionable or do not directly relate to our research question. 

For example, as the reviewer points out, we cannot change the study design. Moreover, our research question was unrelated to political party affiliation-- and there is no reason to believe that party affiliation would impact decision self-efficacy.

Additionally, we did not measure childhood vaccines other than influenza. Influenza vaccine seemed to us to be the most relevant as it shares many more similarities with COVID-19 vaccines (e.g., respiratory virus, need for boosters) than do other childhood vaccines. 

With regard to parental knowledge about vaccines, there is abundant evidence that knowledge does not predict behavior change. Parents can feel confident about decisions for which they are entirely uninformed. However, based on socio-behavioral theory, those who make decisions with confidence are more likely to follow-through with their decisions.

Finally, the issue of whether parents feel efficacious in other aspects of their lives is not really relevant to our study; self-efficacy, as defined by theory, is related to a specific behavior, and not necessarily generalizable to other behaviors.

The role of self-efficacy is articulated in a number of socio-behavioral theories (eg., Ottawa Decision Support Model,[1] Health Belief Model,[2] Theory of Reasoned Action)[3] and with prior research in the context of other health behaviors.[4] Given that decision self-efficacy is a potentially modifiable target for intervention and has not been previously studied, we believe that our paper makes a unique and valuable contribution to the literature.

Literature cited

[1] O’Connor AM, Tugwell P, Wells GA, et al. Randomized trial of a portable, self-administered decision aid for postmenopausal women considering long-term preventive hormone therapy. Med Decis Mak Int J Soc Med Decis Mak. 1998;18(3):295-303.

[2] Janz NK, Becker MH. The Health Belief Model: A Decade Later. Health Education Quarterly. 1984;11(1):1-47.

[3] Ajzen, I. From intentions to action: A theory of planned behavior. In J. Kuhl & J. Beckman (Eds.), Action control: From cognitions to behaviors. 1985: pp. 11–39. New York: Springer

[4] Stacey D, Légaré F, Lewis K, Barry M, Bennett C, Eden K, Holmes-Rovner M, Llewellyn-Thomas H, Lyddiatt A, Thomson R, Trevena L. Decision aids for people facing health treatment or screening decisions. Cochrane Database Syst Rev. 2017 Apr 12;4:CD001431. doi: 10.1002/14651858.CD001431.pub5

Reviewer 3 Report

All suggested edits and comments were addressed sufficiently. 

Author Response

Thank you for the feedback you provided in your review. It has improved the manuscript.